# Immunogenicity of Endolysin PlyC

**DOI:** 10.3390/antibiotics11070966

**Published:** 2022-07-18

**Authors:** Marek Adam Harhala, Katarzyna Gembara, Daniel C. Nelson, Paulina Miernikiewicz, Krystyna Dąbrowska

**Affiliations:** 1Laboratory of Phage Molecular Biology, Department of Phage Therapy, Hirszfeld Institute of Immunology and Experimental Therapy, Polish Academy of Sciences, 53-114 Wroclaw, Poland; marek.harhala@hirszfeld.pl (M.A.H.); katarzyna.gembara@hirszfeld.pl (K.G.); paulina.miernikiewicz@hirszfeld.pl (P.M.); 2Research & Development Center, Regional Specialist Hospital, 53-114 Wroclaw, Poland; 3Institute for Bioscience and Biotechnology Research, University of Maryland, Rockville, MD 20853, USA; nelson@umd.edu

**Keywords:** endolysins, antibodies, deimmunization, Pal, Cpl-1, epitopes, cross-reactions

## Abstract

Endolysins are bacteriolytic enzymes derived from bacteriophages. They represent an alternative to antibiotics, since they are not susceptible to conventional antimicrobial resistance mechanisms. Since non-human proteins are efficient inducers of specific immune responses, including the IgG response or the development of an allergic response mediated by IgE, we evaluated the general immunogenicity of the highly active antibacterial enzyme, PlyC, in a human population and in a mouse model. The study includes the identification of molecular epitopes of PlyC. The overall assessment of potential hypersensitivity to this protein and PlyC-specific IgE testing was also conducted in mice. PlyC induced efficient IgG production in mice, and the molecular analysis revealed that PlyC-specific IgG interacted with four immunogenic regions identified within the PlyCA subunit. In humans, approximately 10% of the population demonstrated IgG reactivity to the PlyCB subunit only, which is attributed to cross-reactions since this was a naïve serum. Of note, in spite of being immunogenic, PlyC induced a normal immune response, without hypersensitivity, since both the animals challenged with PlyC and in the human population PlyC-specific IgE was not detected.

## 1. Introduction

Endolysins are bacteriolytic enzymes derived from bacteriophages that mainly function to lyse the bacterial peptidoglycan for release of progeny phages during a bacteriophage replication cycle. Significantly, purified endolysins are able to hydrolyze the bacterial peptidoglycan, when they are added to sensitive bacteria in the absence of bacteriophages, resulting in the rapid osmotic lysis of the bacterial cell [1,2]. Endolysins alone, thus, represent an alternative to antibiotics, due to their bacteriolytic mechanism. Additionally, since phage endolysins are directly lytic to the peptidoglycan on contact, they are not susceptible to conventional antimicrobial resistance mechanisms, such as efflux pumps [3]. The World Health Organisation has recognised endolysins as innovative, non-traditional biologicals under development [4]. Protein drugs are a rapidly growing part of the health industry. A curated database of FDA-approved therapeutic peptides and proteins, THPdb, suggests there are 239 approved proteins and peptides as well as 380 approved variants of these proteins/peptides [5,6]. Protein enzymes as drugs have important advantages due to their high specificity, proteinaceous nature that excludes chemical toxicity, and high potential for modification and further development [7]. Several companies in the field of biotechnology and pharmacy have been conducting human clinical trials with endolysins. For instance, Contrafect is conducting a phase 3 medical trial on the endolysin exebacase (clinicalTrials.gov, accessed on 21 June 2022, identifier: NCT04160468) and iNtRON Biotechnology completed a phase 2 medical trial of N-Rephasin Sal200 in 2021 (clinicalTrials.gov, accessed on 21 June 2022, identifier: NCT03089697). In February 2020, the FDA validated endolysin technologies as antimicrobial biologics by granting “Breakthrough Therapy” status to the ContraFect endolysin in the phase 3 trial. Further endolysins are currently under pre-clinical development in both industry and academia.

Despite the success, protein therapeutics face a major hurdle: non-human proteins are efficient inducers of specific immune responses, including recognition and neutralization of foreign proteins (e.g., the IgG response) or development of an allergic response, mediating type 1 hypersensitivity (e.g., the IgE response). For these reasons, the immune response may limit the efficacy of non-human protein therapeutics. In particular, after the first exposure to a foreign proteinaceous drug, development of neutralizing IgG antibodies could lead to treatment failure in the subsequent use of the same therapeutic. Antibodies are capable of recognizing foreign antigens by a specific match to epitopes. This specific interaction and binding between IgG and targeted proteins triggers an immune response against the protein, and typically, the protein is neutralized [8,9]. 

Here, we evaluated the general immunogenicity as observed in a human population and in a mouse model, as well as molecular epitopes of PlyC. PlyC is an endolysin originally discovered in the C_1_ bacteriophage and produces immediate “lysis from without” in the absence of the parental bacteriophage against *Streptococci* sp. groups A, C, and E, thus demonstrating the potential for antibacterial prophylaxis. Notably, the antibacterial potency of PlyC endolysin has been demonstrated in vivo, since PlyC protectes mice from colonization by *Streptococci* as well as it decolonizes mice already infected [10]. Moreover, in the cell line infection model of a human, PlyC efficiently removed *Streptococci* from cell cultures, including the successful removal of intracellular bacteria, and these effects were completed in a dose-dependant manner (Appendix A) [11]. Its crystal structure was determined, revealing that the PlyC holoenzyme consists of one copy of the PlyCA subunit (i.e., catalytic domain) and eight copies of the PlyCB subunit (i.e., cell wall-binding domain) [12]. Both the multimeric nature and the inclusion of two gene products are rare in endolysins. PlyCA includes an N-terminal glycosyl hydrolase (GyH) domain, a helical docking domain, C-terminal cysteine, and a histidine-dependent amidohydrolase/peptidase (CHAP) domain. All domains connect by linker regions, and both the GyH and CHAP domains are enzymatically active. PlyCB forms an octameric ring that interacts with PlyCA via its helical docking domain [12,13]. Here, we sought to understand immunogenicity of PlyC, focusing on the PlyC regions/domains that are targeted by specific IgG antibodies. We also included an overall assessment of potential hypersensitivity to this protein by PlyC-specific IgE testing.

## 2. Results

### 2.1. Identification of Immunogenic Epitopes in PlyC by IgG Induction

Baseline immunogenicity was identified in mice that were challenged intraperitoneally with PlyC (100 µg/mouse). Serum levels of PlyC-specific IgG were assessed by ELISA from 1 to 70 days. The enzyme proved to be immunogenic and produce a typical induction of specific IgG, with an increase observed beginning from day 15 with the highest levels observed approximately one month after injection (adj. *p* < 0.05) (Figure 1). 

To identify specific regions that mediate the immunoreactivity of PlyC, we adapted the VirScan methodology, which includes the generation of a phage display library of epitopes, immunoprecipitation, and epitope identification by next-generation sequencing (NGS) (Figure 2) [14]. Briefly, PlyCA and PlyCB subunits were virtually cut into 56 amino acid-long oligopeptides (with 46 amino acid overlap/10 amino acid shift), which were reversely translated, synthesized as oligonucleotides by nucleotide printing and cloned into a T7 phage display library. This library was amplified and immunoprecipitated with PlyC-specific murine sera, and NGS was used to identify sequences of immunoprecipitated epitopes. Significantly increased reactivity with PlyC-specific IgGs was detected in seven oligopeptides (Figure 3a,b), thus allowing for the estimation of four immunogenic regions within PlyCA: 1–9 aa, 91–146 aa, 171–226 aa, and 351–406 aa (Figure 3c). No significant increase of the reactivity was observed within PlyCB. (adj. *p* < 0.01; Figure 3b).

### 2.2. Screening for PlyC-Specific Antibodies in the Healthy Human Population

Serum samples from healthy volunteers (*n* = 56) were tested for reactivity to PlyC by ELISA. Anti-PlyC IgG levels showed significantly higher levels in 12.5% of the samples (7 out of 56) compared to in the overall population (outliers) (Figure 4a). Since the PlyC holoenzyme consists of two subunits, PlyCA (the domain of enzymatic activity) and PlyCB (the binding domain), both domains were individually tested for serum reactivity. Approximately 10% (5 out of 56) of samples showed significantly higher anti-PlyCB IgG levels than the overall population. Noticeably, all five outliers for PlyCB laid within outliers detected for the whole PlyC holoenzyme. The levels of anti-PlyCA showed no outliers. Furthermore, anti-PlyCA IgG levels followed a normal Gaussian distribution, whereas both PlyC and PlyCB significantly skewed from a Gaussian distribution because of overrepresentation of the high reads (Appendix A).

Next, the reactivities of healthy volunteers’ serum IgG for PlyC, PlyCA, and PlyCB were plotted against each other to elucidate any correlations in reactivity within each individual. PlyC values were found to strongly correlate with those observed for PlyCB (Spearman *r* = 0.820; *p* < 0.0001); however, the correlation of PlyC with PlyCA was markedly lower and found to be statistically insignificant (Spearman *r* = 0.24; *p* > 0.01) (Figure 4b,c). Taken together, the data suggested the PlyCB subunit mediates PlyC−IgG interactions in the investigated human sera. 

### 2.3. PlyC-Specific IgE 

IgE antibodies are linked to hypersensitivity, and they can be indicators of allergic reaction to a specified protein targeted by IgE. We, therefore, investigated human sera from a normal population (*n* = 104) for the potential reactivity of the IgE fraction to PlyC. As demonstrated by ELISA, no significant reactivity of IgE was detected against PlyC when compared to the negative control (PBS). Additionally, the overall level of reactivity was much lower than that of the positive reference, which contained purified human IgE (Figure 5). 

The possible induction of IgE was further tested in mice challenged with PlyC. The development of PlyC-specific IgE was assessed by ELISA up to 30 days after challenge. No increase of PlyC-specific IgE was observed, no significant differences to control mice was detected, and the overall detection levels for anti-PlyC IgE were much lower than that of a positive ovalbumin reference in allergic mice (Figure 6). 

## 3. Discussion

Endolysins, as foreign protein enzymes, may induce specific immune response in animals and humans, particularly antibody responses. PlyC is also immunogenic and capable of specific IgG induction as demonstrated here in mice. The highest levels of PlyC-specific IgG was achieved within one month, and the overall profile of induction was typical for many proteins of prokaryotic origin, including those observed for other endolysins (Figure 1) [15]. Molecular analysis revealed that PlyC-specific IgG interacts with particular regions of PlyC in a mouse model. Specifically, four immunogenic regions were identified, all within the PlyCA subunit, whereas none of the IgG fractions appeared to target PlyCB (Figure 3 and Appendix A).

In humans, deliberate exposure to PlyC was not possible; however, putative natural exposure could not be fully excluded. In the tested population (*n* = 56), we found seven individuals with significantly increased (when compared to the overall population, False Discovery Rate 1%) serum IgG reactivity to PlyC and five out of these seven demonstrated increased IgG reactivity to PlyCB (Figure 4a). When correlated to PlyC, the reactivity of PlyCB revealed a strong correlation (Spearman *r* = 0.820; *p* < 0.0001), but PlyCA did not (Spearman *r* = 0.24; *p* > 0.01) (Figure 4b,c and Appendix A). Thus, we concluded that the PlyCB subunit was the region that mediated PlyC−IgG interactions in the investigated human sera.

The identification of immunogenic regions of PlyC in the mouse model revealed all these regions were contained within PlyCA, whereas human sera testing indicated that the IgG reaction was within PlyCB only. Murine and human immune systems are not the same, although highly similar [16]. Differences in the IgG responses to proteins are at most moderated with very few examples of somewhat weaker/stronger or delayed/faster response to a specified antigen, for instance the weaker response to the RBD domain of SARS-CoV-2 spike protein in mice [17]. To the best of our knowledge, proteins that in a response promoting conditions are immunogenic to mice only but not to humans (or vice versa) have not been identified so far. Differences between mice (a laboratory model) and humans (a true population) may, however, contribute to differences observed herein. Intensity of IgG production in any animal or human depends on many factors that can be controlled in an animal model, but they cannot be controlled in a human population only naturally exposed to an antigen (here: PlyC enzyme). These factors may include the dose and dosing schedule, administration route, antigen formulation, accompanying compounds, or others [18]. Of note, the phage display library of oligopeptides in the vast majority represents linear epitopes, while ELISA allows including structural epitopes as well. Thus, it is highly probable that epitopes within PlyCB reactive to human sera were the latter ones. 

These contradictory (between mice and humans) findings may suggest that in human populations naïve to PlyC, there may exist IgG fractions demonstrating some reactivity to PlyC that result from cross-reactions rather than from specific immunization with PlyC. That is, other antigens, non-related to PlyC, may elicit specific IgG production that may cross-react with PlyC. Searches using DALI and VAST did not reveal significant similarities between PlyC and proteins that are common inducers of immune responses in humans. However, the searches did reveal a distant structural similarity between the PlyCB monomer with a functionally uncharacterized *Streptococcus mutans* protein. In addition, some similarities were noted to cellulases, the family of enzymes responsible for degrading the insoluble, polysaccharide cellulose, implicated in food processing [12]. These findings indicated a possible source of antibodies capable for cross-reaction to PlyC in humans. They are also in line with the proposition that epitopes found reactive in humans were conformational in nature, since the similarities to potential cross-inducers were found at the structural level. In mice deliberately challenged with the PlyC holoenzyme, specific linear epitopes were identified as the target for the immune response. Their potential significance for a human immune response will be possible to verify, only when human patients within a clinical trial will be treated with PlyC.

PlyC-specific IgE antibodies were tested as potential indicators of allergic reaction to PlyC, since hypersensitivity might potentially limit or even exclude biologics from therapeutic applications. However, no significant IgE reactivity to PlyC was found in the human population, nor PlyC-challenged mice demonstrated any increase of PlyC-specific IgE fraction. Serum IgE levels in these mice were much lower than those demonstrated in an allergic mouse model where the hypersensitivity to a specified allergen (OVA) was induced by a dedicated adjuvant (Figure 5 and Figure 6).

Of note, IgM induction was not investigated in this study, mostly due to its transient characteristics and the leading role of IgG in immunological memory retention. However, in a regular type of specific immune response induction, a peak of IgM precedes high levels of IgG that is typically detected in blood. Thus, in addition to IgG increase as demonstrated herein, one can expect normal IgM induction and rise within days after challenge with the investigated protein, the most probably with subsequent decrease when IgG achieves its high levels. 

This study demonstrated that PlyC does interact with the mammalian immune system and it may induce immune responses manifesting as PlyC-specific antibodies. We have also, for the first time, isolated immunogenic regions of PlyC, targeted by IgG. This can be potentially useful in protein engineering since optimization of endolysin activity by molecular modifications is being intensively developed, particularly with the goal of therapeutic applicability [19]. It is, however, focused mainly on enzyme−bacterium interactions, not on those with the human or animal host. We proposed the identified immunogenic regions as potential targets for deimmunization and other modifications that may improve protein pharmacokinetics and performance in vivo. 

This study also demonstrated that in spite of being immunogenic, PlyC induces a normal immune response, without hypersensitivity. This was observed both in the animal model, where animals were challenged with PlyC and monitored for possible PlyC-specific IgE production, and in a human population, where PlyC-specific IgE was not detected. Other possible pathways of hypersensitivity development cannot be fully excluded, but specific IgE is a strong indicator of allergic reaction. Moreover, animals were carefully monitored for any negative symptoms, including those indicating hypersensitivity, and no negative symptoms were observed (data not shown). These results strongly support the safety of PlyC as a potential therapeutic enzyme, which can be used to combat difficult bacterial infections. 

## 4. Materials and Methods

### 4.1. PlyC Expression and Purification

PlyC, PlyCA, and PlyCB were expressed as recombinant proteins and isolated as described previously [10,20,21]. In brief, oligonucleotides coding the proteins were cloned into pBAD24 plasmids and expressed in *Escherichia coli* (*E. coli*) BL21 by induction with 0.25% arabinose overnight at 22 °C. Expression cells were then harvested by centrifuge (8000× *g* for 5 min) and washed with 20 mM phosphate buffer (pH 7.0). Cells were then lysed by use of lysozyme (Sigma) and sonication. The insoluble fraction was separated by centrifugation (12,000× *g* for 30 min at 4 °C), and the crude supernatant containing the soluble proteins was used for further purification. The PlyCA subunit was purified by precipitation with 0–20% ammonium sulfate (4 °C; overnight). In contrast, PlyCB and the PlyC holoenzyme were purified by application to a hydroxyapatite column (Bio-Rad, Hercules, CA, USA) with elution in a gradient up to 1 M phosphate buffer. Fractions containing the desired proteins were confirmed by SDS-PAGE, pooled, dialyzed and concentrated (Amicon PM-10 membrane). Next, protein samples were applied to a 26/70 Sephacryl S-200 column and eluted to confirm the complex formations of the PlyC holoenzyme and the PlyCB octamer (Appendix A). Fractions containing desired proteins were confirmed again by SDS-PAGE (see [20] for the gel of the same purification lot used here) and stored in PBS at 4 °C for further use [20]. 

### 4.2. ELISA Test

Levels of antibodies were tested as described previously [21]. Briefly, MaxiSorp flat-bottom 96-well plates (Nunc, Thermo Scientific, Europe) were covered overnight with PlyC, PlyCA, or PlyCB (2 µg/well; 0.1 mL), PBS (blank control), or purified human IgE (positive reference). Wells were washed with PBS and blocked with 1% albumin (Sigma-Aldrich). Diluted serum (1/100 in PBS) was added (100 μL of diluted serum per well). The plate was incubated at 37 °C for 2 h and washed with 0.05% Tween-20 in PBS five times. A diluted detection secondary antibody was applied (100 μL per well): peroxidase-conjugated AffiniPure goat anti-IgE (Jackson ImmunoResearch Laboratories) or peroxidase-conjugated AffiniPure goat anti-IgG (Jackson ImmunoResearch Laboratories). The plate was then incubated for 1 h at room temperature in the dark and then washed 5 times again with 0.05% Tween-20 in PBS. Fifty microliters of TBS (Sigma) were then added to each well and incubated for 20 min. Twenty microliters of 2 N H_2_SO_4_ were added, the absorbance was measured at 450 nm (main reading), and the absorbance at 550 nm (background) was subtracted for each well. Additional wells containing PBS were included as a technical control.

As a reference level of specific IgE antibody induction, an oral mouse allergy model to ovalbumin (OVA) was utilized. Allergy to OVA was induced in mice using a dedicated adjuvant as described [15]. Briefly, the mice were injected subcutaneously with OVA (50 µg/mouse) with Al(OH)_3_ as an adjuvant, promoting the hypersensitivity reaction. The injection was repeated after 14 days. Seven days after the second sensitization, the mice were given 20% Egg White Solution (EWS) (Sigma, Poznan, Poland) in the drinking water. IgE levels specific for OVA were evaluated using ELISA as described above and verified by comparison to control mice injected with PBS instead of OVA. The reference represents the mean value for *n* = 6. 

### 4.3. Identification of the Immunogenic Regions of PlyC by EndoScan

This technique was adapted from Xu at al., who developed a high-throughput method for viral epitope identification; a schema of our adapted approach is presented in Figure 2 [14]. We designed in silico a library of amino acid (aa) sequences (56 aa long) derived from PlyCA and PlyCB sequences, with a 46 aa overlap (i.e., a new oligopeptide starting every 10th amino acid). This library was translated into a DNA oligonucleotide library with codons optimized for *E. coli* [22]. Flanking sequences 5′-GTGTGAATTCTTCTTCTTCT-3′ and 5′-TGAACACCTAAGCTTGTGT-3′ were added to the 5′ and 3′ ends, respectively. This oligonucleotide library was synthesized by nucleotide printing (SurePrint™ technology, Agilent) and cloned into a T7 phage display system with the use of a T7Select 415-1 Cloning Kit (Merck Millipore), according to the manufacturer’s instructions. The resulting phage display library of PlyC-derived epitopes was precipitated overnight with 10% PEG-8000 and 0.5 M NaCl at 0 °C. After centrifugation (for 15 min at 15,000× *g* and 0 °C), the pellet was suspended in Tris-EDTA buffer (pH 8.0). 

The remainder of the protocol was derived, in part, from [14,23,24]. Briefly, the wells of a 96-deep-well plate were blocked with 300 µL of 3% bovine serum albumin in TBST (0.15 M NaCl, 0.05 Tris-HCl, 0.05% Tween-20, pH 7.5) overnight at 4 °C. One microliter (1 µL) of murine sera with confirmed (by ELISA) PlyC-specific antibodies was added to each well, followed by the addition of 0.3 mL of the phage display library of epitopes diluted to approximately 10^5^ fold in Phage Extraction buffer (20 mM Tris-HCl, 100 mM NaCl, pH 8.0 and 6 mM MgSO_4_). Two technical replicates were prepared for each sample. The plate was incubated overnight at 4 °C. Next, 20 μL of magnetic Dynabeads Protein A and 20 μL Dynabeads Protein G (Invitrogen) were added to each well and incubated for an additional 4 h. The beads were washed three times with 300 μL of wash buffer (50 mM Tris-HCl, 150 mM NaCl, 0.1% Triton X-100, pH 7.5) and resuspended in 40 μL of water. This water contained the released immunoprecipitated fraction of the phage display library of PlyC-derived epitopes (still bound to IgG that did not interfere with further steps of the procedure). Each sample was stored at −20 °C until used for further steps. The non-immunoprecipitated phage display library (“input sample”) was also frozen and processed in further steps in the same way as the reference.

To prepare DNA for sequencing, two rounds of PCR were performed with a Phusion Blood Direct PCR Kit (ThermoFisher Scientific, Waltham, MA, USA) without DMSO. In the first round, the following primers were used: forward primer 5′-GCCCTCTGTGTGAATTCT-3′ and reverse primer 5′-GTCACCGACACAAGCTTA-3′, and the phage display library sampled (after immunoprecipitation or the reference) served as a template. The second round of PCR was completed with the use of IDT for Illumina UD Indexes (Illumina) on the first PCR round products as templates. The resulting products served as amplicon sequencing libraries in Illumina Technology. At this point, the NGS reactions were outsourced commercially (Genomed, Warsaw, Poland).

NGS reads were mapped by the bowtie2 software package with the use of the originally synthesized nucleotide library as indexes (local mode) [23]. A number of hits that mapped to each reference sequence (highest score) were counted (count, c).

The signal in each sample was calculated according to Equation (1):(1)$$s_{ijm}=\frac{c_{ijm}}{\sum_{i\in I}c_{ijm}},$$
where s is the signal of *i*-th sequence in *j*-th serum sample and *m*-th technical replicate; c is the number of reads mapped to *i*-th sequence in *m*-th technical replicate of *j*-th serum sample; I is the set of all reference sequences (used as indexes to in mapping by the bowtie2 software).

The vast majority of clones were washed away and were not enriched. If detected at all thus, a zero-inflated negative binomial model was used to evaluate the probability of each signal to be random. Next, all *p*-values of oligopeptides detected in the sample were adjusted for multiple hypotheses (FDR method) [24]. Enrichment (in Figure 3): an average signal in technical replicates of the sample divided by average signal in “input samples” of the same sequence resulted in a signal ratio (enrichment). A series of t-tests evaluated the significance of differences in signals for each oligopeptide separately between the input samples (one group) and the tested sera.

### 4.4. Mice Immunization

C57Bl/6 mice were challenged with 100 µg/mouse of sterile PlyC in PBS on day 0 and day 21. Control mice were injected with an equal amount of PBS only. Murine blood was collected into heprinized tubes from the tail vein. Serum was separated by double centrifugation at 2250× *g* and stored at −20 °C.

### 4.5. Human Sera

Human blood was collected into standard clotting tubes (BD SST II Advance) and left for 1 h at room temperature (RT) to clot, and serum was separated from the clot by centrifugation (15 min, 2000× *g*, RT) and then stored at −20 °C for further use. 

### 4.6. Outlier Detection

Outlier detection (Figure 4) was performed by GraphPadPrism, ROUT method. This method detects outliers when given dataset with a 1% maximum accepted false discovery rate (FDR), showing that the maximum of 1% of detected outliers is falsely determined as outliers [25].

## Figures and Tables

**Figure 1 antibiotics-11-00966-f001:**
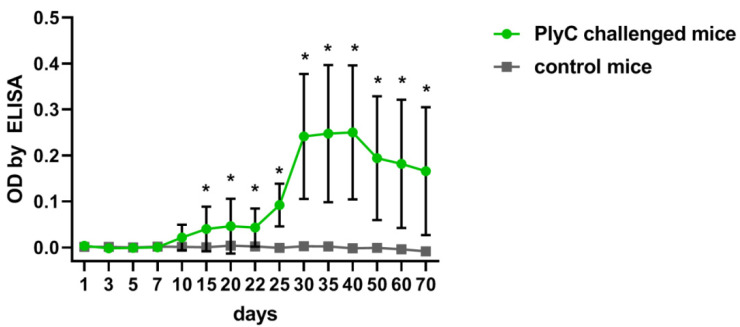
Induction of PlyC-specific IgG antibodies in mice over time. Mice were injected intraperitoneally with PlyC in PBS (100 µg/mouse in 0.2 mL) on day 0 and day 21, and control mice were at the same time injected intraperitoneally with PBS (0.2 mL); serum samples were collected and tested against PlyC by ELISA on indicated days. PlyC-challenged mice were compared to control mice at the same day (*n* = 6). Mean values ± SD are given. *, adj. *p* < 0.05 (Welsch’s t-test). One of two technical replicates is presented.

**Figure 2 antibiotics-11-00966-f002:**
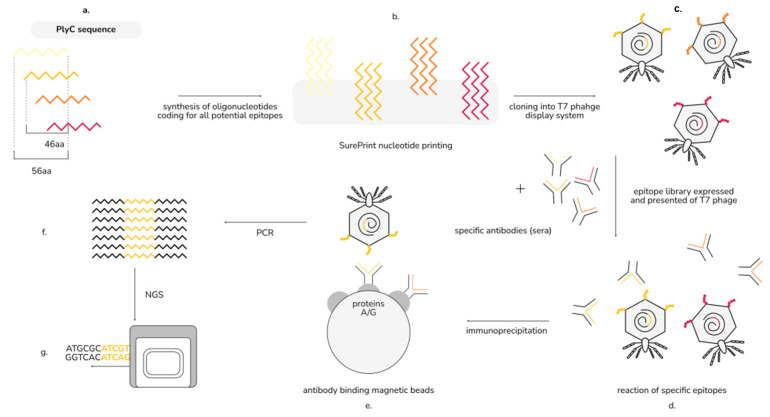
EndoScan technology for the identification of immunogenic regions in PlyC: (**a**) in silico design of peptides covering the sequences of PlyC; (**b**) synthesis of oligonucleotides coding for the peptides; (**c**) construction of a phage display library of PlyC-derived peptides; (**d**) a reaction of the library with specific sera; (**e**) immunoprecipitation with magnetic beads binding Fc fragments of antibodies; (**f**) amplification by PCR reaction; (**g**) next-generation sequencing (NGS) (modified from Xu et al. [14]).

**Figure 3 antibiotics-11-00966-f003:**
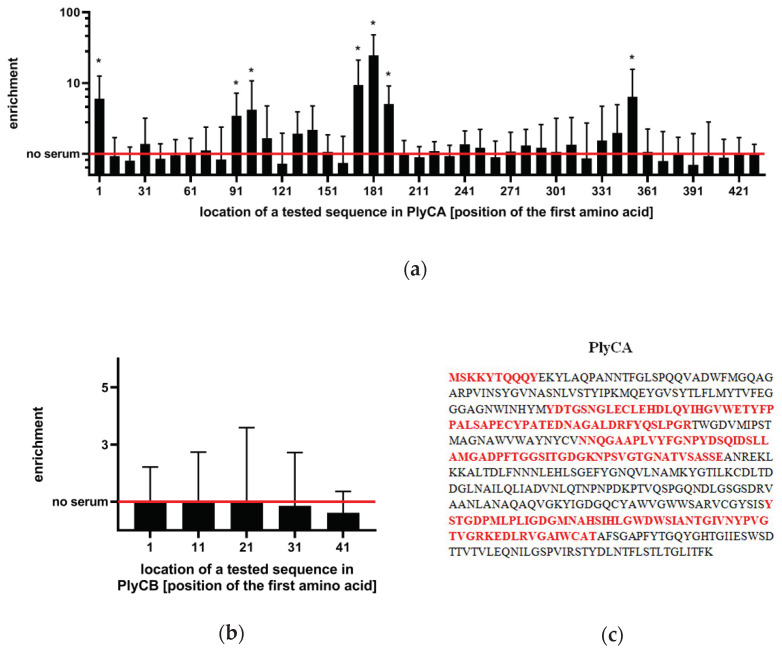
Immunogenic regions in PlyC. (**a**) Enrichment ratios within PlyCA subunit. (**b**) Enrichment ratios within PlyCB subunit. Red lines in panels a and b represent the signal of the ‘input sample’ (that is: the library before immunoprecipitation), normalized as 1.0 (**c**) Estimated immunogenic regions in PlyCA sequence (in red) are presented. The immunogenicity of PlyC regions was defined in the reaction of a phage display library containing PlyC-derived oligopeptides (10 aa shift) with murine sera. Mice were immunized with PlyC (100 µg/mouse in 0.2 mL), and PlyC-specific serum was used for immunoprecipitation. NGS was used to identify the immunoprecipitated fraction of the phage display library. The enrichment ratio of PlyC-representing oligopeptides after immunoprecipitation is presented, indicating the relative strength of interaction. *, adj. *p* < 0.01 from the comparison of normalized signals between the immunoprecipitated sample and the input sample, for each oligopeptide. Datapoints represent the average and the SD of 6 biological replicates (*n* = 6). Two technical replicates were completed.

**Figure 4 antibiotics-11-00966-f004:**
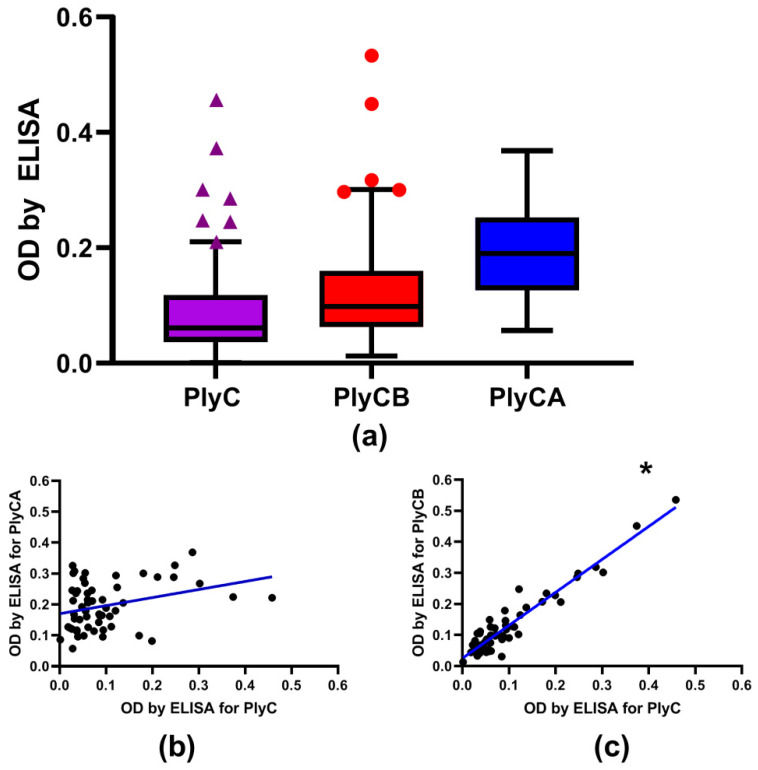
IgG reactivity against PlyC and its subunits, PlyCA and PlyCB, in human sera. Blood sera from healthy volunteers (*n* = 56) were tested against recombinant PlyC, PlyCA, and PlyCB by ELISA. (**a**) Levels of antibodies as ELISA signals; the mean value was presented as the vertical line, 25th and 75th percentile were presented as the box, whiskers designate outliers cut-off, points (triangles or circles) represent outliers in each group, * indicates a *p*-value of <0.0001, and all five outliers that demonstrated the highest levels of anti-PlyCB IgG (circles) and also the highest levels of anti-PlyC IgG’s (triangles). (**b**) Correlation (blue line) between levels of antibodies specific to PlyC and antibodies specific to PlyCA (insignificant). (**c**) Correlation (blue line) between levels of antibodies specific to PlyC and antibodies specific to PlyCB. The correlation between levels of antibodies specific for PlyC and PlyCB was statistically significant (*p* < 0.0001).

**Figure 5 antibiotics-11-00966-f005:**
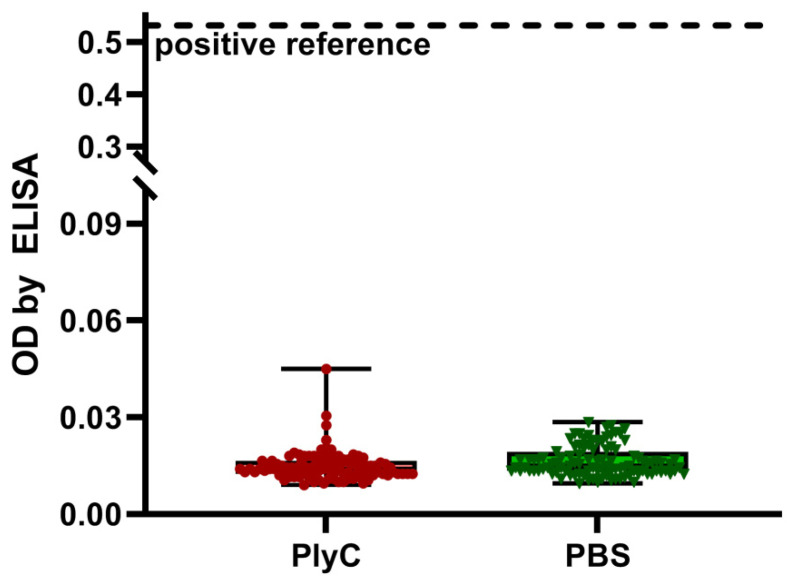
PlyC-specific IgE in a human population. Sera from human participants (*n* = 104) were tested by ELISA for reactivity to PlyC or PBS as a control. A positive reference signal contained wells coated with human IgE. No statistically significant difference was detected between PlyC and PBS reads (*p* > 0.1), despite one sample in the PlyC group determined to be an outlier. Two technical replicates were completed.

**Figure 6 antibiotics-11-00966-f006:**
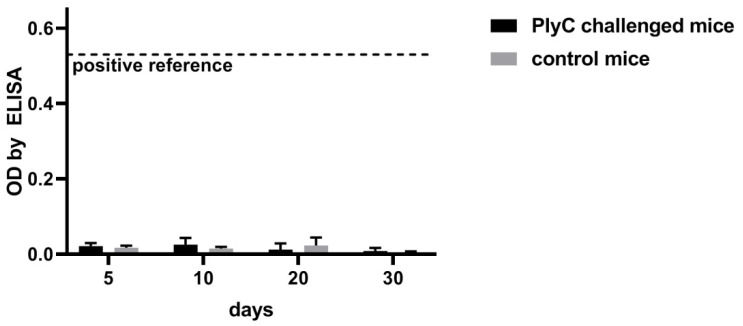
PlyC-specific IgE in mice challenged with PlyC. Mice were injected intraperitoneally with PlyC in PBS (100 µg/mouse in 0.2 mL) or PBS (0.2 mL) control. Serum samples were collected and tested against PlyC by ELISA on indicated days. The positive reference was the average signal to ovalbumin in allergic mice. No statistically significant difference was detected between PlyC-challenged and control mice, but statistical significance was found between the ovalbumin positive reference and all presented groups (adj. *p* < 0.001).

## Data Availability

The data are not publicly available by open access due to privacy and ethical limitations. They can only be partially available on request from the corresponding author.

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
