# Peer review of "Immunogenicity of Endolysin PlyC"

_antibiotics, 2022, doi:10.3390/antibiotics11070966_

Round 1

Reviewer 1 Report

The manuscript by Harhala et al has identified immunogenic regions in PlyC by challenging mice with PlyC. PlyC is a potent endolysin composed of two subunits, PlyCA (Monomer) and PlyCB(octamer). No molecular epitopes were identified in PlyCB when mice were challenged with PlyC and this is interesting because one would expect PlyCB to elicit more response than PlyCA as it’s a octamer and more abundant than PlyCA. When human sera were screened for IgGs that can bind to PlyC, the recovered IgGs bound PlyCB specifically. It would be interesting to see if the authors can recover the PlyCB epitopes using their phage display method in the future. Even though the mice and human study are contradictory, the work presented here is of significant interest and will help in making synthetic versions of PlyC. I only have a few comments that will help to improve the scientific rigor and reproducibility of the manuscript.

Comments

1.      The crystal structure of PlyC (PDB ID: 4F88) is available and highlighting the epitopes on the structure discovered in this study will significantly improve the impact of the manuscript.

2.      Providing a gel for the purified proteins and the Size-exclusion chromatography spectra in the supplemental data will enhance rigor and reproducibility

3.      Revise the methods section for the purification of PlyC, PlyCA and PlyCB with more details on the percentage of ammonium sulfate used for precipitation. Also revise the sentence (Line: 259- 260) as it currently implies that the proteins were purified from inclusion bodies.

4.      In line 316-317 the authors describe that resuspension with water released the T7 phages that were bound to the beads. This is surprising because the interaction between the IgG and the T7 phage displaying the epitope would be very strong.

5.      The following sentences needs revision for clarity.

a.      Line 32 - 35

b.      Line 46 – 48

c.      Line 197 – 200

Author Response

 Please see attachment - a copy is pasted below.

  1. The crystal structure of PlyC (PDB ID: 4F88) is available and highlighting the epitopes on the structure discovered in this study will significantly improve the impact of the manuscript.

Thank you for this comment. We propose a new figure that  highlights the epitopes in the PlyC structure as suggested (Supplementary Figure S1) (Please see attachement).

  1. Providing a gel for the purified proteins and the Size-exclusion chromatography spectra in the supplemental data will enhance rigor and reproducibility

This question is addressed with the longer response to the next question.

  1. Revise the methods section for the purification of PlyC, PlyCA and PlyCB with more details on the percentage of ammonium sulfate used for precipitation. Also revise the sentence (Line: 259- 260) as it currently implies that the proteins were purified from inclusion bodies.

We have revised this section and added additional details for clarity. The purification of these proteins is unique and each requires only a single purification step that does not rely on His tags or other purification tags. PlyCA is purified to homogeneity with a single ammonium sulfate precipitation step of 0-20%. It is the only protein in the E. coli crude extract that precipitates at this low level of ammonium sulfate. The octamer of PlyCB and the PlyC holoenzyme (containing a complex of PlyCA and the PlyCB octamer) bind hydroxyapatite so tightly, that 1 M phosphate buffer is required to elute these proteins, whereas all other proteins elute at <0.5 M phosphate.  Gel filtration is not required for purification, but is done in part to confirm purity/homogeneity of the proteins as well as to verify that PlyB self-assembles into a ~64 kDa octamer and that PlyC self-assembles into the nine-subunit complex containing PlyCA and the PlyCB octamer. All of the above is referenced to prior publications that show gel filtration and SDS-PAGE gels of these proteins. In addition to providing these references and clarifying details in the current manuscript, we now are also providing the individual chromatograms for the gel filtration of the specific purification lot of the proteins used in this manuscript in the new supplemental figure 2. In regards to an SDS-PAGE, the specific lot of proteins used in this manuscript was a large purification lot that was used for multiple experiments in multiple papers. We provide the figure below to satisfy the reviewer’s interest. However, since these same proteins from the same purification lot were also used in other experiments for other papers, this specific image was just published online, so we cannot publish the identical image again. Nonetheless, we now reference this publication and figure to increase the rigor and reproducibility as requested.

The PlyC holoenzyme (~114 kDa; PlyCA subunit = ~50 kDa, PlyCB subunit = ~8 kDa), the PlyCA catalytic subunit (~50 kDa), the C-terminal CHAP domain of PlyCA (18 kDa), and the PlyCB octamer (~64 kDa as an octamer, but displays as an 8 kDa monomer on SDS-PAGE) were purified to near homogeneity as indicated by SDS-PAGE. This image is published as https://doi.org/10.1101/2022.01.06.475266 and therefore will be appropriately referenced, but not duplicate published in the current manuscript. The proteins purified were used for both manuscripts with the exception that the CHAP domain was not used for the current manuscript.

  1. In line 316-317 the authors describe that resuspension with water released the T7 phages that were bound to the beads. This is surprising because the interaction between the IgG and the T7 phage displaying the epitope would be very strong.

This step is according to the manufacturer’s instruction (ThermoFisher Scientific): A low-ionic environment is supposed to break interactions only between IgG and Protein A/G attached to Dynabeads. Thus, interactions between T7 phages and IgG remains unchanged, but antibodies bound to phage capsids do not interfere with further steps in the procedure (PCR). We added this short note to the description to make it more clear.

  1. The following sentences needs revision for clarity
    Line 32 - 35
    Line 46 – 48
    Line 197 – 200

The sentences have been revised as recommended.

Reviewer 2 Report

The authors in this manuscript studied the endolysin immunogenicity of PlyC.

This is an interesting work but there are things that should be improved before publication:

#1- Addition of antimicrobial activity of PlyC or at least a table summarizing all the published;

#2- A schematic figure containing the mechanism of action of analysis will also be very helpul;

#3- Figure 1 - The PlyC challanged mice appears to have in time an increase in  specific IgG, but it will also be very helpful if the authors include the IgM values in their study;

#4- Also missing is the evaluation of antimicrobial response of PlyC challenged mice; 

#5- The authors should also include in their discussion a brief revision on the commerciality of endolysins.

Author Response

Please see attachment - a copy is below.

#1- Addition of antimicrobial activity of PlyC or at least a table summarizing all the published;

In this study, we in fact confirmed previous observations of antimicrobial activity of PlyC. Therefore, we propose the table summarizing abundant previous observations of antimicrobial activity demonstrated for PlyC (Supplementary Table S1).

#2- A schematic figure containing the mechanism of action of analysis will also be very helpul;

Thank you for this suggestion. We agree that the approach should be clarified and we propose the new Figure 2 that includes a graphical schema of the approach.

#3- Figure 1 - The PlyC challanged mice appears to have in time an increase in  specific IgG, but it will also be very helpful if the authors include the IgM values in their study;

We absolutely agree IgM would be very interesting, but we were not able to collect enough blood from animals in this study to secure a sufficient amount for both general screening (IgG increase by ELISA) and epitope identification (by EndoScan). We hope to be able to expand this element in the future.

#4- Also missing is the evaluation of antimicrobial response of PlyC challenged mice;

We agree it would be very interesting to evaluate the antimicrobial response of PlyC in mice, however, due to ethical reasons we were not able to do that: this part of an experiment has not been approved by the Local Ethical Committee. Infections are classified as highly invasive procedures when in vivo. The number of publications on PlyC is constantly growing and we hope that in the close future we will be able to convince the Committee that PlyC is a promising target for extended pre-clinical studies and to get the required approval.

#5- The authors should also include in their discussion a brief revision on the commerciality of endolysins.

We have provided this discussion as part of the revised Introduction by adding the most representative examples that are currently under commercial development and in clinical trials.

Round 2

Reviewer 2 Report

The authors clarified in this manuscript revised version most of the major issues identified in  this study.

This  reviewer agrees with the ethical explanations for suggestion number 4.

In this case, the authors should find a more complex in vitro model to prove their concept (e.g. infection in mice model cells such as L929 cell line).

Also, they should include in the manuscript some sentences about the lack of IgM quantification and its importance.

Author Response

We would like to propose a revised version of our manuscript ‘Immunogenicity of endolysin PlyC’ for your consideration. We are grateful for all comments and advice from the Reviewer and from you, and we did our best efforts to meet the recommendations.

We are grateful for the understanding expressed by the Reviewer, since indeed animal models are more and more difficult to conduct due to formal (ethical) reasons. For these reasons we were not able to include all in vivo elements that we planned, but we hope to be able to extend our study to in vivo in the future, as advised.

Importantly, antibacterial potency of PlyC endolysin in mouse model has been demonstrated in early studies by Daniel Nelson (Nelson et al. Proc Natl Acad Sci U S A 2001;98(7):4107-12), who showed that PlyC protected mice from colonization by Streptococci as well as decolonized mice already infected. As to the cell line infection model for PlyC activity, it has also been reported by the group of Daniel Nelson (Shen et al, eLife. 2016; 5: e13152) who demonstrated the high efficiency of PlyC in removal of Streptococci from cell cultures, in a dose dependant manner, and including intracellular bacteria. For this reason, we propose to refer to these reports in our manuscript giving a better description, and to escape repetitive experiments. We add extended information to Introduction and we hope you will find it good.

As to the Reviewer’s request to include in the manuscript some sentences about the lack of IgM quantification and its importance, we fully agree this will be helpful for a reader and we propose an extended comment added to the Discussion section.